# Proving the Mode of Action of Phytotoxic Phytochemicals

**DOI:** 10.3390/plants9121756

**Published:** 2020-12-11

**Authors:** Stephen O. Duke, Zhiqiang Pan, Joanna Bajsa-Hirschel

**Affiliations:** 1National Center for Natural Products Research, School of Pharmacy, University of Mississippi, Oxford, MS 38655, USA; 2Natural Products Utilization Research Unit, Agricultural Research Service, United States Department of Agriculture, Oxford, MS 38655, USA; Zhiqiang.Pan@usda.gov (Z.P.); joanna.bajsa-hirsche@usda.gov (J.B.-H.)

**Keywords:** allelochemical, mode of action, phytochemical, phytotoxin

## Abstract

Knowledge of the mode of action of an allelochemical can be valuable for several reasons, such as proving and elucidating the role of the compound in nature and evaluating its potential utility as a pesticide. However, discovery of the molecular target site of a natural phytotoxin can be challenging. Because of this, we know little about the molecular targets of relatively few allelochemicals. It is much simpler to describe the secondary effects of these compounds, and, as a result, there is much information about these effects, which usually tell us little about the mode of action. This review describes the many approaches to molecular target site discovery, with an attempt to point out the pitfalls of each approach. Clues from molecular structure, phenotypic effects, physiological effects, omics studies, genetic approaches, and use of artificial intelligence are discussed. All these approaches can be confounded if the phytotoxin has more than one molecular target at similar concentrations or is a prophytotoxin, requiring structural alteration to create an active compound. Unequivocal determination of the molecular target site requires proof of activity on the function of the target protein and proof that a resistant form of the target protein confers resistance to the target organism.

## 1. Introduction

In this review, we define the mode of action (MOA) of a phytotoxin as the process by which it affects a plant, including its primary target site. Understanding the MOA of phytotoxic phytochemicals (PPs) has both academic and practical utility. On the academic side, this knowledge can be useful in the determination of whether a putative allelochemical is actually functioning as an allelochemical [1]. For example, if the MOA of a compound is known from laboratory studies and there is a biological marker for this MOA, one can determine if the marker occurs when target plants are exposed to the levels of the compound found in soil. For example, the putative allelochemical sorgoleone is a potent inhibitor of photosystem II (PSII) [2], for which a rapid increase in variable chlorophyll fluorescence is a biomarker [3]. Dayan et al. [4] found a PSII inhibitor effect on variable fluorescence in *Amaranthus retroflexus* L. grown in the same pot as a sorghum cultivar that secretes the allelochemical sorgoleone into the soil, indicating that sorgoleone is an allelochemical that contributes to the adverse effect of the sorghum cultivar on *A. retroflexus*. This is the only example of this approach to prove the role of an allelochemical of which we are aware, but this method is limited by the little we know of the MOA of allelochemicals and also perhaps by few scientists taking advantage of this approach to prove allelopathy. 

Knowing the MOA of an allelochemical can also be useful in predicting whether weeds will evolve target site resistance to it if a crop is generated that creates a strong selection pressure with that allelochemical. Such an assessment is routinely conducted on new herbicides, as the success of a herbicide product that requires hundreds of millions of US dollars to bring to market can be limited if weeds evolve resistance quickly. Some molecular targets are prone to rapidly evolving resistant forms of the target (e.g., acetolactate synthase (ALS)), whereas others take much longer for target-site resistance to evolve (e.g., 5-enolpyruvylshikimate-3-phosphate synthase (EPSPS)) [5]. 

There is interest in allelochemicals as templates for new classes of herbicides with new MOAs. Herbicides with new MOAs are needed urgently as a tool to manage the rapidly growing evolution of weed resistance to herbicides with the twenty-five or so MOAs currently used by commercial herbicides [5,6]. The triketone herbicides that inhibit hydroxyphenylpyruvate dioxygenase (HPPD) were derived from the triketone phytochemical leptospermone [7] produced by some members of the Myrtaceae family, such as *Callistemon citrinus* and *Leptospermum scoparium*. The triketone herbicides were the last major herbicide group with a new MOA [8]. It is interesting that there is no rigorous study proving that leptospermone and related triketones from these plants act as allelochemicals in nature, although their activity in soil [9,10] supports this conclusion. Moreover, there was no published proof that leptospermone and its analogs from these plants were HPPD inhibitors until after the synthetic triketones were commercialized [11]. There are probably other allelochemicals with MOAs yet to be discovered that could lead to the development of useful new herbicides for which target site resistance does not exist in weed populations.

The determination of the MOA of a phytotoxin is not a trivial exercise, especially if it is a previously unreported MOA. For example, the novel targets sites (serine/threonine protein phosphatases and acyl-ACP thioesterase) of the herbicides endothall [12] and cinmethylin [13], respectively, were not discovered until years after their commercialization. For each of these herbicides that may have been inspired by natural compounds (cantharidin and 1-4-cineole for endothall and cinmethylin, respectively (Figure 1)), there were published missteps before the target sites were clearly discovered. The purpose of this paper is to give general guidance to those who wish to determine the MOA of PPs, an area of research with few good examples of established MOAs. Much of the literature on PPs uses the term allelochemical for these compounds. However, this is usually done without rigorous proof of their role as allelochemicals in plant/plant allelopathy. The fact that a PP is a potent phytotoxin does not prove that it is an allelochemical (e.g., artemisinin [14]), a topic discussed in an earlier publication [1]. Therefore, we use the term PP, regardless of whether or not the compound has been proven to be an allelochemical. 

## 2. Molecular Target Sites and Resulting Secondary and Tertiary Effects

Obtaining an effect of a strong phytotoxin (e.g., a herbicide) on almost any process of a plant is easy. This is one of the reasons that most of the literature on the MOA of PPs describes only secondary and/or tertiary effects of the direct interaction of the PP with a primary molecular target. This is also true for most of the earlier papers on the MOA of synthetic herbicides, although for the past few years, companies have generally discovered and divulged the primary target sites of new herbicides, preventing the publication of papers claiming secondary effects as the MOA in more rigorous journals. 

Papers on secondary and tertiary effects often describe the effects of a PP on a battery of processes and enzymes that are associated with plant stress, particularly caused by the creation of reactive oxygen species (ROS) (e.g., [15,16]). ROS generation is a general effect of stress [17]. Therefore, if a PP is effective, it will cause plant stress, including the creation of ROS. Thus, success in finding such an effect is guaranteed. There are a few herbicides, such as photosystem I energy diverters (e.g., paraquat) and inhibitors of protoporphyrinogen oxidase (PPO) (e.g., acifluorfen), for which the creation of ROS is closely connected to the molecular target [18,19]. Many other papers on MOA seem to be an examination of the effects of the PP on a battery of processes for which the authors were equipped to measure. In the case of effects on an enzyme, these papers usually examine the effects on the amount of enzyme activity extracted from a treated plant, rather than examination of the direct in vitro effect of the phytotoxin on the enzyme. Even when they find a profound effect on extractable enzyme activity, they seldom measure a direct effect on the enzyme in an in vitro assay (e.g., [20]). Again, success in finding effects is almost guaranteed if the PP is effective, especially if the measurements are taken at a time just before or after visual symptoms on the plant occur. However, such papers seldom shed much light on the primary target of a phytotoxin, telling us little about its MOA. 

One could argue that any effects of a PP on a plant are part of the MOA, but this is not helpful in the determination of the molecular target(s) of a PP. The MOAs of commercial herbicides are based on their molecular targets and the secondary effect(s) that are most tied to that molecular target [6]. For example, herbicides that exert their effects by binding to the α-tubulin subunit (e.g., the dinitroaniline herbicides such as trifluralin) are termed mitotic inhibitors because their binding to the α-tubulin subunit protein directly inhibits mitosis, even though other processes are more indirectly affected. Thus, elucidation of the MOA of any phytotoxin should be based on the effect on a primary target.

Rather than determining the effects of a PP in an assortment of assays that are convenient, a more direct approach to finding the MOA of a phytotoxin is to look for phenotypic effects of the compound and focus on processes and molecular targets that might be related to the observed phenotypic effect (see Section 4.3). Since there are many potential molecular targets, this approach will often lead to the researchers learning new assays for potential targets that they did not anticipate. Thus, MOA research is demanding, requiring researchers to either learn new methods or collaborate with those who are experts with the methods indicated by phenotypic or physiological clues.

## 3. Clues from Molecular Structure and In Silico Binding Studies

Similarities between the molecular structure of a compound with an unknown MOA to that of a compound with a known MOA can provide an obvious clue to the MOA. For example, the MOA of the commercial herbicide endothall (Figure 1) was unknown for many years. However, its structural similarity to the natural compound cantharidin (Figure 1), known to inhibit serine/threonine protein phosphatases, led Bajsa et al. [12] to determine whether this was also the case for endothall. It was, and they also found cantharidin to be a potent phytotoxin by the same mechanism. However, a small change in a chemical structure can cause a change in MOA. For example, there are phytotoxic diphenyl ether compounds with very different MOAs, such as PPO (e.g., acifluorfen) [19], acetyl-CoA carboxylase (ACCase) (e.g., diclofop) [21], solanesyl diphosphate synthase (aclonifen) [22], plastid ATPase (nitrofen) [23], and enoyl reductase (the synthetic compounds triclosan and the fungal phytotoxin cyperin) [24] (Figure 2). All of these are target sites for commercial herbicides except for enoyl reductase. Diphenyl ether compounds have been reported from plants (e.g., [25]), but we are unaware of reports of whether any of them are phytotoxic or not. There may be other target sites for other diphenyl ether compounds that have not been described, and it is highly likely that some diphenyl either compounds have more than one molecular target site as a phytotoxin (see Section 4.7). Diphenyl ethers are perhaps an extreme case, but they are a clear example that structural similarity clues can be misleading, even though they are sometimes helpful. 

Structural similarity to a biochemical intermediate can indicate that a compound may interfere with the pathway in which that intermediate is found. For example, the highly potent, microbially produced phytotoxic fumonisins and AAL-toxin are structural analogs of sphingoid bases (e.g., phytosphingosine), intermediates of the ceramide synthesis pathway, and are inhibitors of ceramide synthase in plants and animals [26]. Another example is that of the fungal phytotoxin 2,5-anhydro-d-glucitol (AhG), a close analog of fructose [27]. Like fructose, it is phosphorylated by plant glycolytic kinases to produce AhG-1,6-bisP (Figure 3), which inhibits fructose-1,6-bis P aldolase, its molecular target as a phytotoxin. 

A more modern approach in using the molecular structure of a phytotoxin to determine its molecular target site is to use computational chemistry to determine whether it tightly binds a plant enzyme for which the crystal structure is known using the methods of McRobb et al. [28]. This approach, along with transcriptome data (see below), was used to determine that the PP citral probably exerts its phytotoxicity effects by the inhibition of single strand DNA-binding proteins [29]. In this case, this analysis indicated that the PP binds to several such transcription factors, accounting for the transcriptome results. In silico binding studies for PPs are limited by the relatively small proportion of plant enzymes for which the crystal structure is available; however, we expect that the utility of this approach will increase substantially as computing power and the number of elucidated plant protein crystal structures increases. 

Molecular structure clues, whether based on computational chemistry binding data or otherwise, like phenotypic, physiological, and biochemistry clues, must be verified by direct studies of the effects of the phytotoxin on the proposed target protein(s) (see Section 4.6).

## 4. Finding the Primary Molecular Target with Physiology and Biochemistry

Use of physiological and biochemical methods for identifying the molecular target site of a PP is usually essential. The methods below give a brief synopsis of such approaches, along with their pitfalls and difficulties. 

### 4.1. Choosing Test Organisms 

A test plant species that is genetically uniform and sensitive to the phytotoxin is important. Furthermore, a species for which there is considerable genetic information is desirable for reasons discussed in Section 4.5 and Section 5. Thus, even though weeds are the target of herbicides, they are generally not suitable for most MOA studies, as the sensitivity to a phytotoxin within a population can vary considerably because of non-uniform genetics. Moreover, little or no information is available on the genomes of most weeds. Despite considerable progress in next generation sequencing techniques, including sequencing quantity and quality, as well as shortening the duration of the process, a shortage of annotated genomes is still a substantial hindrance. Available raw data of transcriptomes or genomes of a variety of plant species, downloaded by numerous researchers into Sequence Read Archive, a part of National Center for Biotechnology Information database (https://www.ncbi.nlm.nih.gov/sra), can help. The protein coding sequence or amino acid sequence of interest can be recovered by blast and alignment of reads to a homologous sequence. 

Small-seeded cultivated crop or grass species, such as lettuce (Lactuca sativa) and bentgrass (Agrostis stolonifera), are ideal dicot and monocot species, respectively, for initial studies, as they lend themselves to microbioassays when there is very little test compound for study, as is often the case with natural compounds [30]. Lemna species (duckweeds, e.g., Lemna paucicostata) are also good because they are small and genetically uniform due to asexual reproduction, and their two-dimensional growth habit lends itself to the non-destructive analysis of growth effects with image analysis [31]. Other species are ideal for particular bioassays, such as cucumber (Cucumis sativa) for the measurement of rapid effects on plasma membrane integrity (e.g., [32]). However, parts of plants without functioning chloroplasts or mitotic cells (e.g., grass coleoptiles, [33]) are not appropriate for primary screens for phytotoxicity because all of the potential molecular targets are not functioning in such plant tissues. 

Initial bioassays are useful in the determination of potency and the general phenotypic effects of a phytotoxin (see Section 4.3). However, because of its well-studied genome and biochemistry, we recommend that *Arabidopsis thaliana* (Arabidopsis) be used for the more definitive genetic studies described below (Section 5) that can unequivocally prove the target site of a phytotoxin. It is also a very small plant that is ideal for microbioassays that require small amounts of test material. Although not as two-dimensional as duckweed, image analysis can also be used for the growth analysis of Arabidopsis. 

### 4.2. Achieving the Optimal Dose and Timing 

A complete dose/response curve for the PP is essential information for initial MOA studies. This will allow the selection of an appropriate dose (concentration) of the phytotoxin for further studies. The response chosen is generally a growth parameter such as plant weight, longitudinal growth, or leaf area. If the concentration used is too high, the chances of affecting more than one target site are increased, complicating the MOA study (see Section 4.6). Moreover, secondary and tertiary effects will occur much more rapidly at high concentrations, making identification of the primary target more difficult. We find the most useful dose/response curves are generated with half log doses (e.g., 1, 3.3, 10, 33, etc., mM) versus the measured effect because there is generally a linear response of quantitative parameters (e.g., growth) to log doses [33]. After a complete dose/response curve is generated, we recommend selecting a dose at or near that causing the maximum effect, as there will be less variation in the response at this concentration. Two compounds with the same MOAs should have similar dose/response curve slopes, although there are exceptions [34]. 

Furthermore, the dose/response curve will vary with the time after exposure to the PP, so this should be considered when moving from dose/response studies to physiological, biochemical, and genetic studies. Changes in the dose/response relationship can be due to many factors, including the rate of uptake of the toxicant, its half-life in the organism, and the cascade of secondary and tertiary effects after inhibition of the primary target protein. In general, the longer the time after exposure, the more complicated the effects on the plant become. Thus, determination of the molecular target site is generally easier at a time point soon after that target comes into contact with a strong dose of the PP. This time point is not always simple to identify unless there is a biomarker that is easily identified, such as increases in variable chlorophyll fluorescence for PSII inhibition [3] or rapid increases in shikimate accumulation for inhibition of EPSPS [35]. Another variable that can affect the dose/response curve is the proportion of the target site that must be inhibited to achieve an effect. This varies considerably, with some targets such as PPO requiring relatively little of the target site to be inhibited [36].

### 4.3. Visible Phenotypic Response Clues

Phytotoxins with the same MOA will generally cause the same visible phenotypic effects. For example, inhibitors of mitosis, such as synthetic dinitroaniline herbicides (e.g., trifluralin) and natural inhibitors like colchicine, cause root tip swelling [37]. Auxin mimic phytotoxins (e.g., 2,4-D) cause rapid growth, epinasty, and shoot twisting and turning [18]. Although compounds with the same MOA cause particular symptoms, compounds with a different MOA can often cause very similar symptoms. For example, the production of unpigmented (white) foliage is the result of inhibited carotenoid production. This phenotype can be due to the inhibition of a number of enzymes in the carotenoid pathway, as well as the inhibition of the synthesis of plastoquinone (PQ), a needed cofactor for phytoene desaturase (PDS), a key enzyme of the carotenoid synthesis, by inhibition of any of the enzymes leading from tyrosine to PQ (Figure 4). Nevertheless, such a profound phenotype narrows the target site to a relatively few potential molecular targets. A profile of phenotypic responses to phytotoxins with known MOAs in a standard species can be generated to obtain clues as to the MOA of a compound for which the MOA is unknown. This approach has been used by a company in its search for herbicides with new MOAs. They combined visible phenotypic effects and a few physiological measurements on several plant species to build profiles for this purpose, terming their approach physionomics [38]. However, even if a compound fits the profile for a known MOA, the molecular target must be proven by genetics and/or in vitro target site assays. If the phenotypic profile of a compound does not fit that of any known mode of action, it may have a new molecular target and MOA. 

### 4.4. Reversion and Inhibitor Studies

One of the simpler ways to probe the MOA of a PP is to determine how exogenously fed metabolites, inhibitors, or other compounds that affect plant metabolism affect the phytotoxicity of the PP. For example, if a phytotoxin inhibits an enzyme in a biochemical pathway, supplying a metabolite or metabolites produced by that pathway can reduce or eliminate the effect of the PP. This type of experiment is called a reversion, supplementation, or metabolic rescue study [30]. Table 1 provides examples of this approach. A good dose/response curve is required for this type of experiment, as the optimal phytotoxin dose for reversal is the lowest dose for almost complete inhibition. Doses of the phytotoxin that are much higher than this may be affecting targets other than the main phytotoxin target, the effects of which would not be affected by the putative reversing chemical(s). Moreover, a dose/response experiment should be conducted with the chemical(s) used for reversal, to make sure that the dose/concentration used is not phytotoxic. 

In addition to ameliorating the effects of phytotoxins by providing substrates or products of the target enzyme (Table 1), supplemental chemicals can reduce the effects of any toxic compounds produced by the action of the phytotoxin. For example, antioxidants can reduce the effects of phytotoxins that produce large amounts of ROS. For example, the strong antioxidant pterostilbene reduces the effect of the PPO inhibitor herbicide aciflorfen [48]. Some compounds can reduce the effect of a phytotoxin by chemically reacting with it. For example, the PP dehydrozaluzanin C is apparently partly phytotoxic by covalently binding thiols of proteins and other biomolecules [49]. Its phytotoxicity was greatly reduced by supplying the oxidized form of glutathione, to which the PP covalently bound, inactivating it. However, supplying histidine and glycine provided 40% reversal of the effects of this PP, indicating that effects other than binding thiols contribute to its phytotoxicity. If the MOA of a phytotoxin involves the accumulation of a toxic intermediate, administering a compound that blocks the pathway to that intermediate can reduce the effects of a phytotoxin. For example, providing gabaculine (a product of *Streptomyces toyacaensis*) to plant tissues treated with synthetic herbicide inhibitors of PPO prevents the accumulation of highly toxic protoporphyrin IX by the inhibition of the synthesis of aminolevulinic acid, a precursor for the porphyrin pathway [50,51]. Another inhibitor of the porphyrin pathway (dioxoheptanoate) also reverses the effects of PPO inhibitors [50]. These results were useful in the identification of PPO as the target of this class of herbicides [52]. 

Reversion studies can sometimes be misleading. For example, Grossmann et al. [53] used a combination of physionomics and reversion studies to conclude that cinmethylin is an inhibitor of tyrosine aminotransferase. However, in vitro inhibition of the enzyme was weak, and later studies, using a chemoproteomic approach, found cinmethylin (Figure 1) to act by the inhibition of fatty acid thioesterases to stop the production of medium length (C14-C18) fatty acids, a target unique to this herbicide [13]. In our study on the mode of action of *t*-chalcone [42], feeding plants homogentisate reversed the effects of this PP, indicating that it inhibited the synthesis of PQ by the inhibition of HPPD, but no effect was found on this enzyme in an in vitro assay. Sometimes, reversion studies give only a weak effect, but their results can accurately indicate the biosynthetic pathway affected. For example, some reversion studies using aromatic amino acids to reverse the effects of glyphosate gave weak results, but later in vitro enzymology and genetics research proved unequivocally that this very successful herbicide inhibits only one enzyme, EPSPS, an essential enzyme for the shikimate pathway that produces aromatic amino acids [54].

### 4.5. Omics Methods

Omics methods (e.g., transcriptomics, proteomics, metabolomics, and physionomics) are potentially useful tools in determining the target sites of phytotoxins. However, it is usually very difficult to find a clearly defined target in the large amount of data usually generated by these technologies, as has been discussed in two reviews [55,56]. In fact, there is no good example of the use of these technologies alone leading to the discovery of the MOA of a phytotoxin, although there are many papers that use these methods. There are several difficulties. One of these is that all phytotoxins cause stress, and a large number of genes are upregulated by stress. For example, in an attempt to probe the mode of action of the PP allelochemical benzoxazolin-2,(3H)-one (BOA) [57], the phytotoxin was found to upregulate a large number of stress- and xenobiotic-related genes that made the determination of a primary target site impossible. This paper found similar effects of an assortment of other phytotoxins (e.g., 2,4-D) and xenobiotics (e.g., phenobarbital). As mentioned above, others have often mistaken stress effects for the primary effects of phytotoxins. Stress effects are seen at every genetic, biochemical, and physiological level. 

Use of the software that arranges the data according to biochemical pathways, such as the Kyoto Encyclopedia of Genes and Genomes (KEGG), can clarify the data considerably. For example, KEGG analysis of transcriptome data from a study examining the effects of the PP t-chalcone on Arabidopsis narrowed the target site to only a few metabolic pathways, with only one consistent with the phenotypic effects, the pathway from tyrosine to plastoquinone (PQ) [43] (Figure 5). The results were the same for both roots and shoots. This approach suggested that PQ synthesis was impaired, and reversion studies with homogentisate (a PQ precursor) (Table 1) with both duckweed and Arabidopsis supported this, indicating that HPPD was the target. However, there was no in vitro inhibition of the activity of the enzyme. The authors speculated that t-chalcone could be converted to a HPPD inhibitor in vivo. In vivo bioactivation has been documented for natural products (see Section 4.8). 

In another attempt to find a target site of a PP, we examined the effect of citral on the transcriptome of Arabidopsis [29]. This was more successful, in that there was a rapid downregulation of approximately a third of the genome in roots of Arabidopsis within 1 h of exposure. This made pinpointing a metabolic target impossible, but as described above, we found by computational chemistry methods to determine protein binding that citral is likely to be a good inhibitor of single strand DNA-binding proteins that are transcription factors. This MOA explained the unusual transcriptome effects well. We also found the natural phytotoxin cantharidin to affect the transcription of a large number of genes of Arabidopsis [58], which was to be expected because we found cantharidin, as well as its herbicide analogue endothall, to both inhibit all of the several classes of serine/threonine protein phosphatases in this plant [12]. 

Analysis of transcriptome effects when we know the molecular target of a toxin shows that there is not always a strong effect on the expression of the gene encoding the target. For example, analysis of the transcriptome effects of a group of fungicides that are known ergosterol synthesis inhibitors found that genes for enzymes other than that of the molecular target in yeast were more affected; however, all of these ergosterol synthesis inhibitors produced a similar effect on the pattern of expression of all of the ergosterol pathway genes [59]. Zhu et al. [60] found no transcription effect by glyphosate on the gene for its target, EPSPS. Thus, with even the most accurate and sophisticated transcriptome analysis, this procedure is not likely to locate a clear target among the many genes of the plant. However, it does produce an accurate account of the effect of the phytotoxin on gene expression. 

The target site of almost all phytotoxins is a protein. Nevertheless, proteomics results can be even more problematic in the interpretation of results because of the many posttranslational effects on proteins. As a result, transcriptomic effects are not always reflected in proteomics effects, indicating that changes in the proteome are not solely the consequence of changes in the transcriptome [61,62,63]. In a rare experiment in which transcriptome (microarrays) [58] and proteome (two-dimensional difference in gel electrophoresis method (2D-DIGE)) [64] data were taken from the same tissues in the same experiment, there was almost no correlation between these two omic profiles in Arabidopsis in response to cantharidin, other than the upregulation of glutathione-S-transferases involved in detoxification of xenobiotics. Part of the poor correlations could also be due to the fact that proteomics methods are not as definitive with low abundance proteins as modern transcriptomics methods with low abundance mRNA. Relatively new and more sensitive alternatives for 2-D or 2D-DIGE are, e.g., Q-Orbitrap mass analyzers that enable global proteome profiling and quantitative protein analysis [65]. Despite the fact that this method has been available for a couple of years, articles covering the discussed issues in this review have not yet emerged. Low abundance protein targets are highly desirable for herbicides because they are more likely to need less herbicide to kill the plant [36], but finding such targets with proteomics is problematic. Protein binding interactions (not enzymatic) with potential inhibitors when many proteins are examined at the same time have been termed chemoproteomics [66,67]. This method was used by Campe et al. [13] to identify the molecular target of cinmethylin and by Counihan et al. [68] to identify the thiolase enzymes involved in fatty acid oxidation that are targeted by the herbicide acetochlor in mice livers. This method is especially useful to identify targets for which in vitro enzyme assays are not available or are very difficult. 

Metabolomics has been used by one company involved in new herbicide discovery to find potential herbicides with new molecular targets [53,69]. Metabolomics involves monitoring the effects on a smaller number of compounds than of genes or proteins. However, there are few clear metabolite biomarkers for most phytotoxins. Exceptions are shikimate and quinate for glyphosate, protoporphyrin IX for PPO inhibitors, and sphingoid bases for inhibitors of plant ceramide synthase [35]. In these cases, the biomarker compounds are generally in very low abundance in untreated cells of most plant species because of their phytotoxicity. Most biochemical metabolites are not phytotoxic and are found at much higher concentrations in plant tissues, making changes in their concentrations caused by a phytotoxin less obvious than the marked changes seen in low abundant metabolites such as protoporphyrin IX or shikimate. Thus, changes in metabolite profiles caused by inhibitors of many of the enzymes of primary metabolism may not be sufficiently profound to provide a clear clue to the MOA. For example, changes in the metabolome of duckweed caused by the aglycone of auscaulitoxin indicated that alanine aminotransferase was the target enzyme, but in vitro assays of the effects on the activity of this enzyme found no effect [46]. Clearly, this phytotoxin affects some aspect of amino acid metabolism, as evidenced by a complete reversal of its effects by feeding the treated plants with several amino acids individually, but metabolomic data could not identify the target. Furthermore, the effects were reversed by so many amino acids that a single pathway was not indicated. The authors speculated that it affects one or more amino acid transporters, a target that would be virtually impossible to identify by the effects on metabolite profiles. Metabolomics has been used to probe the effects of phytotoxic doses of the PPs biochanin and catechin on Arabidopsis, but there were no clear indications of their MOAs [70]. Likewise, metabolome effects of the PP allelochemical umbelliferone on durum wheat suggested a primary effect on aromatic amino acid metabolism but did not determine the target site(s) [71]. A successful use of metabolite analysis to determine the MOA of a phytotoxic sugar (7-deoxy-sedoheptulose—see Table 1) from a cyanobacterium found that it caused the accumulation of 3-deoxy-D-arabino-heptulosonate 7-phosphate in treated plants [47]. This compound is a substrate of an early enzyme of the shikimate pathway, 3-dehydroquinate synthase, and the authors found it to inhibit this enzyme. 

As with all MOA studies, results will vary with time after treatment and dose of the phytotoxin. The speed of the effect will also vary with the time for movement of the phytotoxin to the target site and sometimes with the speed with which the plant can bioactivate (see Section 4.8) or detoxify the compound. The earlier in the cascade of effects that can produce a measurable effect, the more likely the effect will be to the primary target. Much of the omics literature dealing with the effects of a toxin is confounded with too many effects to see what the target might be.

Another issue is that some phytotoxins target mainly certain tissues or cell types, e.g., inhibitors of mitosis. Moreover, in a whole plant, the phytotoxin concentration may vary considerably between cell types, tissues, and organs. Thus, extraction of mRNA, proteins or metabolites from whole plants, shoots, or roots may dilute an effect that is found mostly in target cell types. Methods are now available for microsampling from cells and tissue, especially for transcriptomics (e.g., [72]), but without already knowing the MOA, knowing what cells to sample is problematic. 

Ordinary omics methods measure pool sizes of mRNAs, proteins, or metabolites at a single point in time. However, most of these pools are in a state of flux, with input and losses. Changes in pool size do not necessarily reflect changes in turnover within the pool. For example, when dark-grown maize seedlings are placed in the light, the phenylalanine pool flux rate is increased more than three-fold, even though the pool size is reduced by 40% [73]. The work was done with old-fashioned pulse-chase experiments, using ^14^C-phenylalanine, HPLC, and a radioactivity detector. Metabolomics can now be done with pulsed stable isotopes and chromatography, followed by mass spectrometry, a procedure called fluxomics [74]. Such methods have been used with phytotoxins. For example, glyphosate was found to increase de novo amino acid synthesis in a weed with this method [75]. 

As before [55,56], we still cannot point to any omics study that has yielded the clue pointing to a verified MOA of a herbicide or a natural phytotoxin. Even when more than one omics method has been used, the results have been disappointing. For example, Araniti et al. [76,77] used both proteomic and metabolomic methods, as well as physiological methods, to study the effect of the PPs coumarin and thymol on Arabidopsis, respectively, but no clear indication of a definitive MOA was found for either compound. Artificial intelligence methods can be useful to predict MOAs from large omics databases [22] (see Section 6).

### 4.6. Direct Measurement of Effects of the PP on Molecular Targets

One of the two clearest ways to ascertain the target of a phytotoxin is to demonstrate that it inhibits the function of that target in an in vitro assay at a low concentration. The other is to show that a resistant form of that target that is not affected by the phytotoxin renders the plant resistant to the phytotoxin (see Section 5.2). Physiological assays for various known herbicide targets are available in books (e.g., [78]) and reviews (e.g., [79]) and many research papers. All of the methods described before this section can point to a target, but the in vitro assay is critical for proof of a MOA. Unfortunately, clear indications of a target are often found to be wrong by the in vitro bioassay (e.g., [43,46]. However, in other cases, indications are verified by an in vitro assay (e.g., [12]). The in vitro bioassay of a putative molecular target is not infallible. For example, reversion and metabolomic results indicated that cinmethylin (Figure 1) acted on tyrosine aminotransferase, and a weak in vitro effect on the enzyme seemed to verify this [53]; however, later studies found that it acts on acyl-ACP thioesterase in lipid metabolism [13]. A potential difficulty in verifying physiological or omics clues with an in vitro assay on a protein is that the phytotoxin may require in vivo chemical modification to bind the protein target. This topic is dealt with in Section 4.8. 

Direct determination of the effects on a protein is the step in MOA discovery that is missing from most MOA publications, especially those on natural product phytotoxins. This is probably because many enzymes are not easily assayed (e.g., EPSPS and PPO). This is a primary reason why good MOA research requires hard work in learning new methods and/or collaboration with experts who have the skills needed for the indicated target site.

### 4.7. Complications by Multiple Targets

All commercial herbicides appear to have a single molecular target, except perhaps some of the natural product herbicides like pelargonic acid (discussed in [80]). However, natural product phytotoxins often appear to act at more than one molecular target. For example, the allelochemical sorgoleone can inhibit PSII, mitochondrial electron transport, HPPD, and H^+^-ATPase [81]. Another example of a PP phytotoxin with more than one MOA is that of sarmentine, a compound from Piper species being developed as a natural phytotoxin [82]. It inhibits both PSII and enoyl-ACP reductase at low concentrations. An advantage in nature of more than one molecular target is that this will reduce the probability of the evolution of resistance at the target site [83]. However, more than one target makes the determination of the MOAs difficult, especially if the importance of the different MOAs varies with environmental conditions, species, and dose. Nevertheless, the different MOAs can be identified with sufficient effort (e.g., [81,82]). The relative importance of different molecular targets can be made clearer by mutants with resistant forms of the target protein (Section 5.2).

### 4.8. Complications by Prophytotoxins

Some phytotoxic molecules that are synthesized by humans or biosynthesized by living organisms are inactive at the molecular target and must be metabolically converted to the active molecule. Such molecules are termed prophytotoxins. There are numerous examples of proherbicides [84]. There are two advantages of proherbicides. First, the physicochemical properties of the proherbicide may be better for movement to the target site than the actual enzyme inhibitor. Secondly, selectivity of the proherbicide can be influenced by whether a plant species can activate it or not. Examples of prophytotoxins in nature are the microbial compounds hydantocidin that must be phosphorylated in the plant to be an inhibitor of its target, adenylosuccinate synthase [42], and 2,4-anhydro-d-glucitol that must be glucosylated to inhibit its target, fructose-1,6-bis P aldolase [27] (Figure 3). Based on biochemical and physiological factors, we speculated that the PP t-chalcone is a prophytotoxin [43]. In some cases, the prophytotoxin might be converted to the enzyme inhibitor in the environment, rather than in the target plant. For example, the weakly phytotoxic allelochemical and relatively unstable PP benzoxalinone, 4-hydroxy-2H-1,4-benzoxazin-3(4H)-one (DIBOA), can be converted to the much more stable and phytotoxic aminophenoxazine, 2-amino-3H-phenoxazin-3-one (APO), with the weak phytotoxin benzoxazoline-2(3H)-one (BOA) as an intermediate, in the soil (Figure 6), potentially confounding allelopathy or MOA studies [85]. APO was found to be a strong inhibitor of histone deacetylase, whereas DIBOA and BOA were inactive on this enzyme [86]. Thus, the confounding influence of metabolic activation can be overcome if one knows the active derivative of the prophytotoxin.

## 5. Genetic Proof of the Target

### 5.1. Comparing Phenotypes of Known Mutants with That of Treatment with the Phytotoxin

A similar phenotype of a characterized mutant can provide a clue as to the MOA. However, since potent phytotoxins kill plants, a mutation of the target to cause an effect similar to the toxin may be lethal. Chimera plants can have mutations in some tissues that would be lethal if found in all of the plant cells. For example, a chimera with albino patches in the leaf was found to lack the plastid enzyme polyphenol oxidase in the chlorotic tissues [87]. This provided a clue that the uniform, similar phenotype caused by the potent cyclic tetrapeptide produced by a plant pathogen, tentoxin, might also lack this enzyme. Tentoxin-treated seedlings completely lacked the enzyme [87]. The ultrastructure of the plastids of the mutant and those affected by tentoxin were similar. Thus, lack of this enzyme was connected to the MOA of tentoxin. 

### 5.2. Generating Genes for Resistant Targets

The most definitive method for proving that a phytotoxin has a particular molecular target is to generate a plant that has this target that is resistant to the inhibitor at the in vitro level. For example, plants containing transgenes with resistant forms of EPSPS have resistance levels to glyphosate that are around 50-fold greater than untransformed, isogenic plants [88]. If there were another significant target, this would not be possible. There are similar results with transgenic and mutant plants for many other commercial herbicides, indicating that the herbicides to which they have high levels of resistance have only one significant target [89]. Similarly, a transgene for resistance of dihydroxy-acid hydratase, an enzyme of the branched chain amino acid pathway, to the fungal phytotoxin aspterric acid proved this enzyme to be the unequivocal target of this fungal phytotoxin [90].

### 5.3. Genetics Approach Using Resistant Mutants

As mentioned above, the model plant Arabidopsis can be used to identify specific molecular target sites for PPs. This can be achieved by screens for genetic resistance or hypersensitivity in Arabidopsis, an approach known as forward genetics. The genomic loci responsible for the herbicide or phytotoxin resistance are first identified by screening ethyl methanesulfonate (EMS)-mutagenized Arabidopsis seeds, and the gene responsible for the phenotype is then isolated using conventional genetic mapping-based cloning or genomic DNA sequencing. The publicly available T-DNA lines or complementation by overexpression of the target gene in the mutant lines can then be used for further proving the function of the target genes for the herbicides/phytotoxins. An example of such an approach was the case of isoxaben and thiazolidinone herbicides, which were reported to inhibit cellulose synthesis. Two semidominant mutants ixr-1 and ixr-2 were isolated from EMS-mutagenized Arabidopsis that confer resistance to these herbicides [91,92]. Later, the locus was mapped to the Arabidopsis genome using genetic mapping, and the mutated gene was identified as cellulose synthase, a molecular target of isoxaben and thiazolidinone [93]. Another example is the result of screening EMS-mutagenized Arabidopsis using sulfamethoxazole, a compound that belongs to the antimicrobial sulfanilamide family, which is phytotoxic to plants. A mutant was isolated with reduced sensitivity to sulfamethoxazole, and the gene responsible for the phenotype was identified encoding 5-oxoprolinase using genomic mapping and whole-genome sequencing [94]. Although this forward genetic screen has proven a useful tool in identifying the molecular targets of bioactive compounds, it is limited in cases in which the molecular target is an essential gene, which may cause lethality when mutated. Nevertheless, this approach provides a tool for the discovery of MOAs and proving the molecular target sites of phytotoxins.

## 6. Artificial Intelligence in MOA Discovery—Prospects

Existing databases with information about chemical structures, their bioactivity in different bioassays, and genomic, transcriptomic, proteomic, and metabolomic data, etc., can be used by researchers using artificial intelligence (AI) methods. Machine learning is often used synonymously with AI, but it is one of the methods/tools of AI. AI is a collection of computational tools utilizing sophisticated statistical software that can analyze massive datasets to find correlations, learn, and subsequently utilize these relationships in resolving specific tasks [95]. This area of science has recently dramatically expanded and progressed. The largest pharmaceutical companies have partnered with well-established or startup companies that offer such advanced approaches in drug discovery and development [96]. The exploitation of AI has the potential to shorten the time and costs of the discovery and development of lead compounds [97]. The agrochemical industry is also using AI in its pesticide and pesticide MOA discovery efforts [98,99]. AI and machine learning can be useful in the determination of the MOAs of compounds as well as in the discovery of new compounds with known MOAs [100,101]. AI enables at least two different approaches associated with phytotoxin MOAs: 1. designing compounds for a specific MOA with the most optimal weed-specific target and avoiding the high throughput screen of thousands of random compounds, followed by an extensive investigation of their MOA; and 2. determining the MOA of known allelochemicals and other natural phytotoxins by the analysis of large data resources on compounds with known MOAs and comparing with similar data on the compound with an unknown MOA. A rare example of the effective application of the latter approach is an exhaustive investigation of the MOA of aclonifen (Figure 2) [22]. Originally, the MOA of this diphenyl ether herbicide was described as both a putative PDS (Figure 4) and PPO inhibitor [102]. Analysis of RNA-seq transcriptomic results obtained from treatments of Arabidopsis seedlings with 49 herbicides with 40 well-characterized molecular targets was employed to train a random forest MOA prediction classifier (a form of AI). This method reduced the possible impacted biosynthetic pathways by aclonifen to carotenoid biosynthesis. Together with this machine learning approach and application of a wide array of molecular biology and biochemistry methods, the MOA of aclonifen was determined to be by the inhibition of solanesyl diphosphate synthase, an enzyme required for PQ synthesis, which is a required cofactor for PDS, a key enzyme in carotenoid synthesis (Figure 4). We expect further use of AI for MOA discovery in the future as this tool becomes more robust and access to it expands. 

## 7. Summary

This short review gives the reader a conception of the potential complexity of mode of action (MOA) research, whether on phytochemical phytotoxins or other phytotoxins. This complexity accounts for the fact that relatively few research publications on the MOA of phytotoxins have actually discovered the MOA. Most are on the secondary effects of the phytotoxin that the authors can easily determine. In a few cases, one of the clues that we have discussed has given astute researchers the path to determination of the MOA. However, in most cases, especially when a compound has a rare or previously undescribed MOA, the determination of the MOA is challenging, even for herbicide discovery companies with extensive resources. For those with access to large databases on the effects of compounds with known MOAs, artificial intelligence methods of MOA determination will be valuable tools in the future. We hope that the strategies and approaches that we discuss here will help those involved in the determination of the MOA of PPs to achieve more successful outcomes. 

## Figures and Tables

**Figure 1 plants-09-01756-f001:**
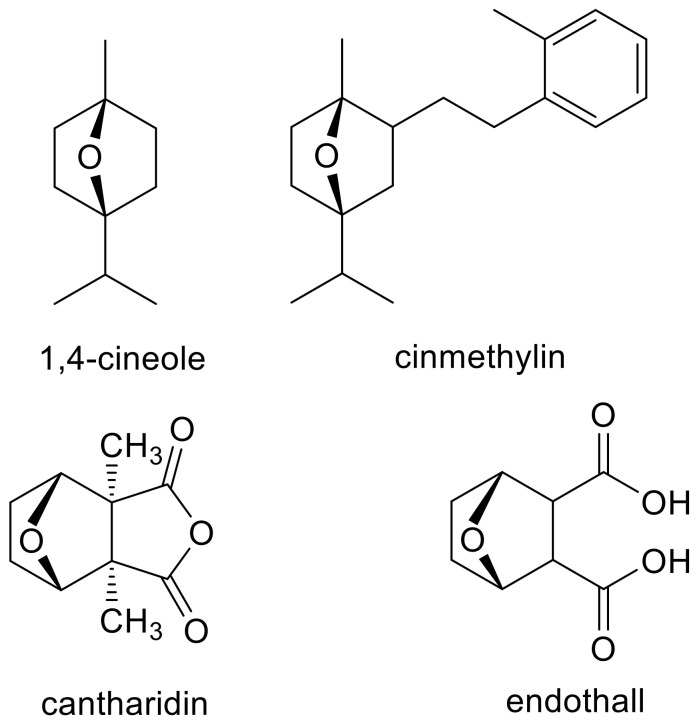
Examples of natural phytotoxins (1,4-cineole and cantharidin) that may have inspired synthetic, commercial herbicides (cinmethylin and endothall).

**Figure 2 plants-09-01756-f002:**
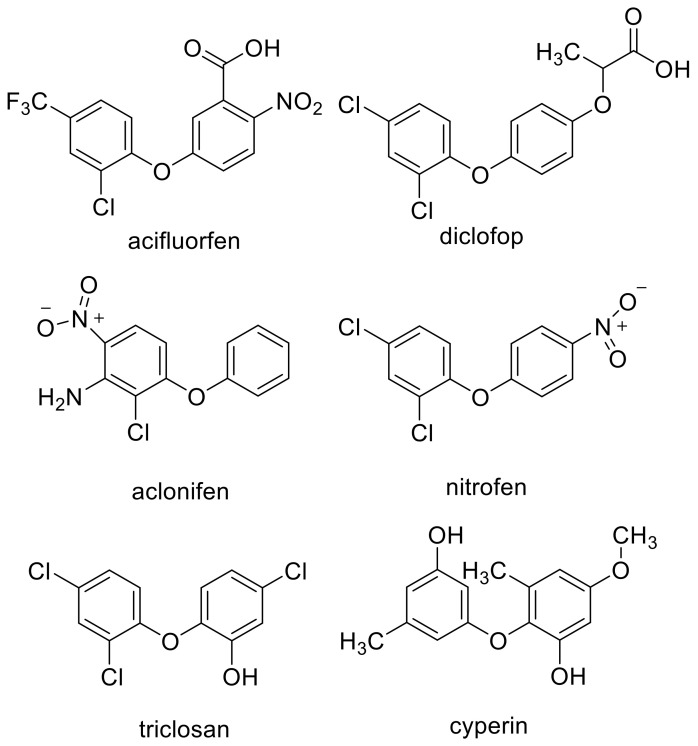
Examples of diphenyl ether herbicides and other phytotoxins with similar structures but with different modes of action.

**Figure 3 plants-09-01756-f003:**
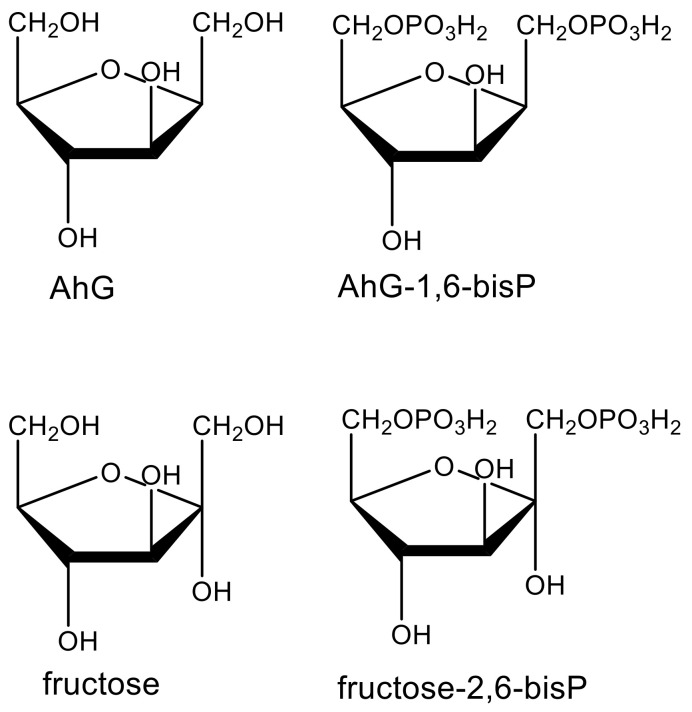
Structures of the fungal phytotoxin 2,5-anhydro-d-glucitol (AhG), fructose, and their phosphorylated forms.

**Figure 4 plants-09-01756-f004:**
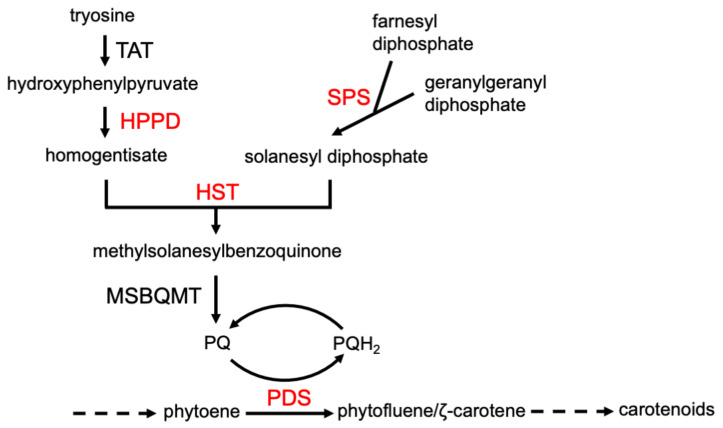
Some of the enzymes (largest font) required for synthesis of carotenoids, the inhibition of which causes a white phenotype. Those in red are targets of synthetic herbicides. TAT = tyrosine amino transferase, HPPD = hydroxyphenylpyruvate dioxygenase, HST = homogentisate solanesyltransferase, SPS = solanesyl diphosphate synthase, PDS = phytoene desaturase, MSBQMT = methylsolanylbenzoquinone methyl transferase. Plastoquinone (PQ) is a PDS cofactor.

**Figure 5 plants-09-01756-f005:**
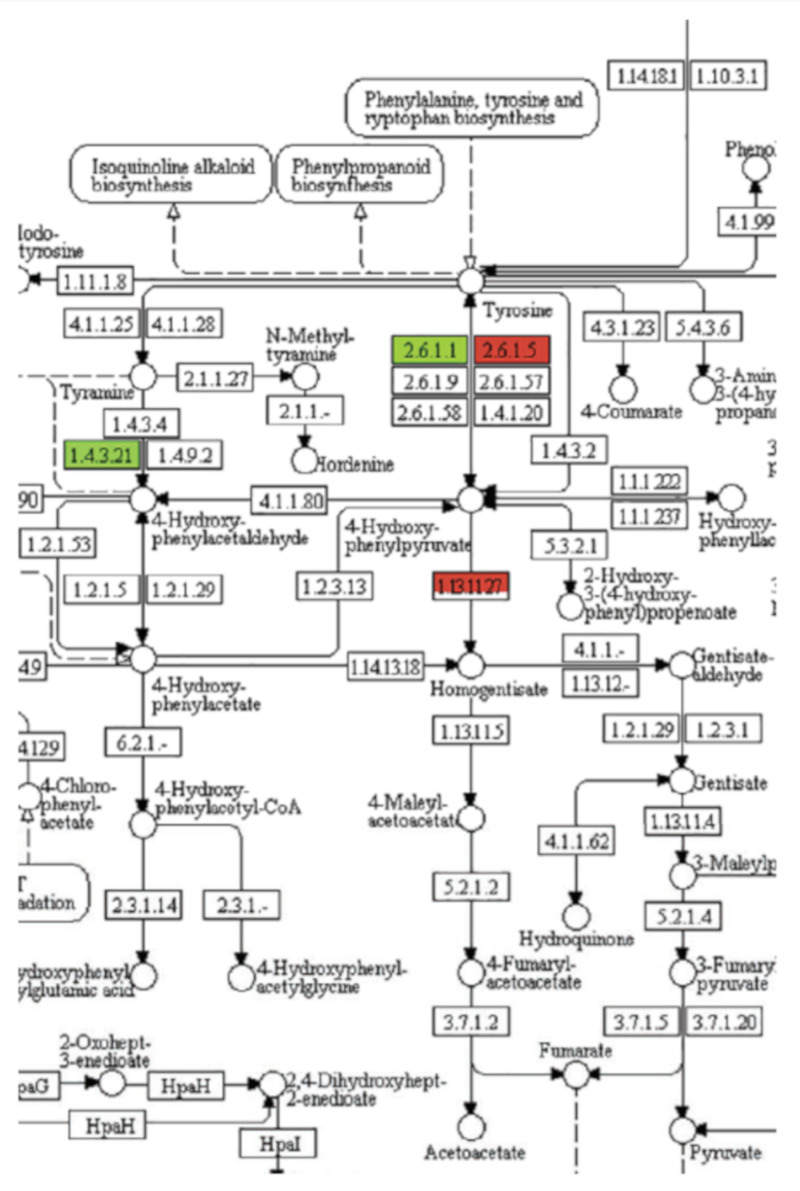
Example of Kyoto Encyclopedia of Genes and Genomes (KEGG) analysis of the effects of *t*-chalcone on transcription of genes of tyrosine metabolism of *Arabidopsis* roots 1 h after exposure. The level of color saturation of green and red represents the level of down- and upregulation, respectively. This is a portion of a more detailed figure from [43].

**Figure 6 plants-09-01756-f006:**
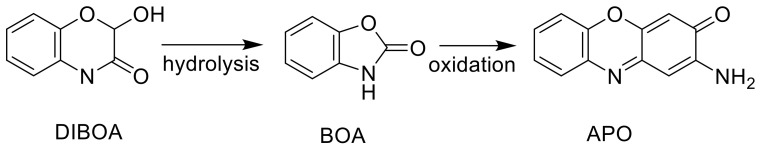
Conversion of the relatively weak and unstable allelochemical (DIBOA) to a strong, more stable phytotoxin (APO) that occurs in soil.

**Table 1 plants-09-01756-t001:** Examples of reversion of effects of both phytotoxic phytotoxins (PPs) and other phytotoxins.

Phytotoxin	Chemical(s) Used for Reversal	Actual or Possible Physiological Process or Enzyme Involved	Ref.
Glyphosate ^1^	aromatic amino acids	EPSPS of shikimate pathway	[39]
Imidazolinones ^1^	branched chain amino acids	acetolactate synthase	[40]
Cornexistin ^2^	aspartate	aspartate aminotransferase	[41]
Hydantocidin ^2^	AMP	adenylosuccinate synthetase	[42]
*t*-chalcone ^3^	homogentisate	plastoquinone synthesis	[43]
Rhizobitoxin ^2^	methionine	β-cystathionase	[44]
Asulam ^1^	folate or *p*-aminobenzoate	7,8-dihydropteroate synthetase	[45]
auscaulitoxin aglycone ^2^	most amino acids	amino acid transporter	[46]
*7-*deoxy-sedoheptulose ^2^	aromatic amino acids	3-dehydroquinate synthase	[47]

^1^ Commercial herbicide, ^2^ Microbially produced phytotoxin, ^3^ PP.

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
