# Peer review of "Proving the Mode of Action of Phytotoxic Phytochemicals"

_plants, 2020, doi:10.3390/plants9121756_

Round 1

Reviewer 1 Report

The  review entitled ‘Proving the mode of action of phytotoxic phytochemicals’ presents information about selected phytotoxic phytochemicals including their structure characterization, in silico binding studies, physiology and biochemistry and a lot of more interesting studies. On the whole, the paper is well considered and informative.

Authors presented the part of science in detail what caused that the paper is long and at one point hard to read. In my opinion, scientists used every efforts to create innovative and interesting review but I think that there is too much information. The presented information are very helpful for scientists interested in this part of natural compounds nevertheless I suggest shortening the paper.

Additionally, I would enriched the paper in more figures and tables which are more readable for scientists and allow to focus attention on concrete issue.

Author Response

Thanks for the kind words. We have tried to improve the prose to make it more readable in places. As requested, we have added more figures. We think that it is actually short review of an enormous topic. 

Reviewer 2 Report

Dear Authors,

The present study aims to review the current approaches to molecular target site discovery of phytotoxic phytochemicals (PPs) and provide general directions to researchers who are interested in determining the mode of action of PPs. The study is in the focus of aims and scopes of Plants and it can add significantly to the current knowledge since less attention has been paid to this area of research. In addition, there are only a few good examples of established modes of action for PPs, as mentioned in the manuscript.

Regarding the importance of the study, relying solely on chemical weed management can be unsustainable both to the environmental impact of herbicides and their residues and insufficient when weed populations evolve resistance to herbicides. Thus, alternative compounds, such as PPs, should be thoroughly investigated and well-understood before integrating them into weed management systems.

Finally, regarding the content and the structure of the manuscript, this short review is well-organized, with an adequate structure. The English level is totally understandable, and the quality of presentation is high. Perhaps you could reconsider the use of “we” throughout the manuscript. We normally suggest using passive voice instead.

Kind regards,

The Reviewer

Author Response

Thanks for the kind words.

This reviewer considered it a short review, whereas the first thought it was too long. So, the length is a compromise. 

We checked to several times for spelling and syntax errors.

Reviewer 3 Report

The authors present a well-written, clearly structured and thematically important manuscript with high degree of originality. From the reviewer's point of view, there is only one simple but helpful petitesse that should be considered.

The use of abbreviations is generally discouraged, especially but not exclusively in the Summary paragraph, which should be readable without any references to the text. Most abbreviations are not necessary anhow, as the terms do normally not occur too often. Confining abbreviations to the absolutely necessary will inprove readybility and, thus, impact of the text considerably.  

Author Response

Thanks for the kind words.

The summary is now readable without reference to the text. We removed some of the abbreviations.